# Age-Standardized Breast Cancer Detection Rates of Breast Cancer Screening Program by Age Group in Korea; Comparison with Age-Standardized Incidence Rates from the Korea Central Cancer Registry

**DOI:** 10.3390/healthcare8020132

**Published:** 2020-05-11

**Authors:** Miye Kim, Hyeongsu Kim, Heejung Choi, MiSeon Son, Kun-Sei Lee, Tae-Hwa Han, Sollip Kim

**Affiliations:** 1Department of Nursing, Graduate School, Konkuk University, Chungju 27478, Korea; evanesel@naver.com (M.K.); mi-sun626@hanmail.net (M.S.); 2Department of Preventive Medicine, School of Medicine, Konkuk University, Seoul 05029, Korea; kunsei.lee@kku.ac.kr; 3Health-IT Center, College of Medicine, Yonsei University, Seoul 03722, Korea; taehwa.han@gmail.com; 4Department of Laboratory Medicine, Inje University, Ilsan Paik Hospital, Goyang 10380, Korea; lalacopine@gmail.com

**Keywords:** breast cancer, screening, incidence rate, detection rate

## Abstract

To compare the epidemiological characteristics of a breast cancer screening program of patients between 40–69 years of age and ≥70 years of age, we calculated the age-standardized detection rate of the breast cancer screening program and compared it with the age-standardized incidence rate from the Korea Central Cancer Registry. The data of the breast cancer screening program from January 2009 to December 2016 and the data of the health insurance claims from January 2006 to August 2017 were used. In the 40–69 year age group, the age-standardized detection rate of breast cancer increased annually from 106.1 in 2009 to 158.6 in 2015 and did not differ from the age-standardized incidence rate. In the ≥70 year age group, the age-standardized detection rate of breast cancer increased annually from 65.7 in 2009 to 120.3 in 2015 and was 1.9 to 2.7 fold of the age-standardized incidence rate. It shows that the early detection effect of breast cancer screening was greater for patients over 70 years old. Further studies are needed to evaluate the effect of breast cancer detection in the ≥70 year age group on all-cause mortality or breast cancer mortality.

## 1. Introduction

Affecting 1.7 million women each year, breast cancer is the most common women’s cancer worldwide. Furthermore, it is also the only type of women’s cancer where its rates are increasing [1]. Recently, Korea discovered a significant change in breast cancer incidence and death aspects due to the rapid westernization of Korean diets over the last 20 years [2]. According to the Korean cancer statistics, breast cancer is the second most common cancer after thyroid cancer, with the incidence and mortality rate continuing to rise [3]. That is, the age-standardized incidence rate of breast cancer in Korea rose from 14.5 per 100,000 women in 1993 to 62.5 in 2016, and the age-standardized death rate from breast cancer increased from 3.7 per 100,000 women in 1993 to 7.0 in 2017 [4,5,6,7]. Age-specific incidences showed peaks in women in their forties. This is causing a steady increase in the individual and social cost burden associated with breast cancer treatment [8].

To reduce the social costs associated with cancer, many countries have adopted screening test-programs which promote early detection and early treatment. Mammography screening is one of the screening test-programs which is specific to breast cancer [9]. Despite the limitation of false positive diagnosis, many studies have reported that mammography screening has reduced breast cancer mortality rates, as early diagnosis has allowed efficient treatment and prevention [10,11]. Consequently, in 1999 Korea also began conducting breast cancer screening on women aged 40 or older every two years [9]. Recently, in terms of compatibility with mammography screening, each country issued recommendations that include the age and cycle recommendations and the level of evidence for breast cancer screening [12,13,14,15]. The National Cancer Center in Korea also announced a revision of the new breast cancer screening recommendations that include age and cycle [6]. The proposed revisions are categorized according to age and should be conducted every two years (recommendation grade B) for women aged 40–69 and asymptomatic. In the case of 70 years of age or older, it is recommended to selectively perform breast cancer screening using mammography (recommendation grade C) in consideration of individual risk and evaluation [9]. In the development process of new breast cancer screening recommendations, only one domestic paper was used [16]. In addition, there have been few studies dealing with the difference in the effectiveness of breast cancer screening programs by age group, used in the recommendation.

In this study, we aimed to compare the differences in epidemiological characteristics on breast cancer screening programs between two age groups. Especially, the age-standardized breast cancer detection rate of the breast cancer screening program was compared with the age-standardized incidence rate from the Korea Central Cancer Registry (KCCR) in order to discuss how we have to consider age factor, in view of the trend of detection rate, in determining the upper age limit for breast cancer screening.

## 2. Materials and Methods

### 2.1. Data Sources and Study Design

In this study, two data set of the National Health Insurance Service (NHIS) were used; (1) The data of the breast cancer screening program from January 2009 to December 2016 and (2) the data of health insurance claims from January 2006 to August 2017. This study used a secondary data analysis to compare cancer detections within 6 months from the time of screening by dividing the breast cancer screening participants into the age groups of 40–69 years and 70 years and older.

### 2.2. Study Subjects

The subjects screening results were classified as normal, benign lesion, suspected breast cancer, and deferred. Suspected and deferred breast cancer diagnoses were defined as positive screening outcomes. The subjects were limited to participants in breast cancer screening not diagnosed with breast cancer. Screening participants who made insurance claims for the code for breast cancer (C50) and the code for special case (V193) from 1 January 2006, until the day of breast cancer screening were excluded. The code for special case aimed to ameliorate the economic burden on patients with cancer, cerebrovascular or cardiovascular disease, severe burns, or other intractable disease by reducing the copayment for 5 years [17].

### 2.3. Variables

The subjects’ identification, age, screening date, result notification date, and screening results were obtained from the breast cancer screening data and the subjects’ identification, sex, breast cancer diagnostic code, code for special case, and dates of visits to medical institutions from the insurance claims data. The subjects were categorized according to age at the time of screening into the 40–69 year age group or the ≥70 year age group. The results of breast cancer screening were classified as ‘normal’, ‘benign lesion’, or ‘suspected breast cancer’. A finding of ‘normal’ or ‘benign lesion’ was categorized as a negative result; a finding of ‘suspected breast cancer’ was considered a positive result. A diagnosis of breast cancer was determined for subjects assigned the code for malignant breast cancer (C50 in the Korean Standard Classification of Diseases) and that for cancer special cases (V193). Therefore, the date of diagnosis of breast cancer was defined as that of the first claim for a code of V193 and C50.

### 2.4. Indicators

The participation rate, positivity rate, positive predictive value (PPV), and detection rate of breast cancer screening were evaluated. 

#### 2.4.1. Participation Rate of Screening for Breast Cancer

The participation rate of screening for breast cancer was defined as the number of breast cancer screening participants who underwent mammography for breast cancer during a particular year per 100 of all women over 40 years old of breast cancer screening subjects.
(1)$$\mathrm{Participation}\;\mathrm{rate}=\frac{\mathrm{Breast}\;\mathrm{cancer}\;\mathrm{screening}\;\mathrm{participants}}{\mathrm{Breast}\;\mathrm{cancer}\;\mathrm{screening}\;\mathrm{subjects}}\times 100$$

#### 2.4.2. Positivity Rate of Screening for Breast Cancer

The positivity rate of screening for breast cancer was defined as the number of positive tests per 100 breast cancer screening participants.
(2)$$\mathrm{Positivity}\;\mathrm{rate}=\frac{\mathrm{Tested}\;\mathrm{positive}}{\mathrm{Breast}\;\mathrm{cancer}\;\mathrm{screening}\;\mathrm{participants}}\times 100$$

#### 2.4.3. PPV of Screening for Breast Cancer

The PPV is the proportion of subjects diagnosed with a disease among those who tested positive for that disease. In this study, the PPV was defined as patients with insurance claims (C50 + V193) for breast cancer codes within 180 days of screening notification among those with positive results of breast cancer screening per 1000 who tested positive.
(3)$$\mathrm{PPV}=\frac{\mathrm{Detected}\;\mathrm{breast}\;\mathrm{cancer}\;\left(C50\;+\;V193\right)\;\mathrm{within}\;180\;\mathrm{days}}{\mathrm{Tested}\;\mathrm{positive}}\times 1000$$

#### 2.4.4. Detection Rate of Screening for Breast Cancer

Among the participants in breast cancer screening, the definition was detection by breast cancer screening if treatment for breast cancer (C50 + V193) occurred within 180 days from the notification date of the breast cancer screening result. Thus, the detection rate of screening for breast cancer was defined as the number of participants in breast cancer screening who tested positive for breast cancer and were treated within 180 days of their notification date per 1000 breast cancer screening participants.
(4)$$\mathrm{Detection}\;\mathrm{rate}=\frac{\mathrm{Tested}\;\mathrm{positive}\;\mathrm{and}\;\mathrm{breast}\;\mathrm{cancer}\;\mathrm{detected}\;\left(C50\;+\;V193\right)\;\mathrm{within}\;180\;\mathrm{days}}{\mathrm{Breast}\;\mathrm{cancer}\;\mathrm{screening}\;\mathrm{participants}}\times 1000$$

To compare the detection rate of the breast cancer screening program with the incidence rate from KCCR of the 40–69 year age group and ≥70 year age group, the detection rate and incidence rate were standardized with the 2009 population in Korea.

Age-standardized detection rate. Females >40 years of age in 2009 who participated in breast cancer screening were defined as the standardized population for calculation of the annual age-standardized detection rate of breast cancer (unit, 10 years).
(5)$$\mathrm{Age}-\mathrm{standardized}\;\mathrm{detection}\;\mathrm{rate}=\sum \frac{\mathrm{Detection}\;\mathrm{rate}\;\mathrm{for}\;\mathrm{age}\;\mathrm{group}\times \mathrm{standardized}\;\mathrm{population}\;\mathrm{for}\;\mathrm{age}\;\mathrm{group}}{\mathrm{Standardized}\;\mathrm{population}}$$

Age-standardized incidence rate. The age-standardized incidence rate of breast cancer according to the KCCR after 2009 (unit, 10 years) was calculated using the number of women >40 years of age in the middle of 2009 according to the National Statistics Office as the standardized population.
(6)$$\mathrm{Age}-\mathrm{standardized}\;\mathrm{incidence}\;\mathrm{rate}=\sum \frac{\mathrm{Incidence}\;\mathrm{rate}\;\mathrm{for}\;\mathrm{age}\;\mathrm{group}\times \mathrm{standardized}\;\mathrm{population}\;\mathrm{for}\;\mathrm{age}\;\mathrm{group}}{\mathrm{Standardized}\;\mathrm{population}}$$

### 2.5. Statistical Analyses

SAS (ver. 9.1; SAS Institute, Cary, NC, USA) statistical software and Microsoft Excel (ver. 2013, Microsoft, Redmond, WA, USA) were used for data analysis. The significance of differences in the PPV and the rate of detection of breast cancer between the two groups was evaluated by calculating the 95% confidence internal (CI).

### 2.6. Ethics Statement

This study used secondary data provided by the National Health Insurance Service (NHIS-2019-1-115) and was exempted from review by the Institutional Review Board of Konkuk University because no personal information was analyzed (approval number: 7001355-201901-E-086).

## 3. Results

### 3.1. Subjects and Participants; and Rates of Participation, Positivity

In the 40–69 year age group, the number of subjects and participants in breast cancer screening and the rate of participation fluctuated from 2009 to 2016 (Table 1). Similarly, in the ≥70 year age group, the number of subjects and participants in breast cancer screening and the rate of participation fluctuated from 2009 to 2016; however, the number of participants increased annually after 2009 (Table 1). The participation rate increased from 52.3% in 2009 to 67.7% in 2016 in the 40–69 year age group and increased from 39.8% to 51.6% in the ≥70 year age group. The rate of participation in the ≥70 year age group was lower than that in the 40–69 year age group from 2009 to 2016. The positivity rate of screening test in the 40–69 year age group decreased from 18.7% in 2009 to 15.2% in 2016 and that in the ≥70 year age group increased from 5.9% in 2009 to 8.0% in 2016.

### 3.2. PPV and Detection Rate 

The PPV of the ≥70 year age group was significantly higher than that of the 40–69 year age group from 2009 to 2016 (Figure 1). The PPV of both groups increased for 2009 to 2016, 40–69 year age group, 5.7 to 10.9 (95% CI, 5.48–5.90 to 10.36–11.46); ≥70 year age group, 11.1 to 17.3 (95% CI, 9.72–12.47 to 16.47–18.21).

The detection rate of breast cancer in the group increased from 1.06 (95% CI, 1.01–1.11) in 2009 to 1.66 (95% CI, 1.58–1.75) in 2016 (Figure 2). The rate of detection in the ≥70 year age group increased from 0.66 (95% CI, 0.62–0.71) in 2009 to 1.38 (95% CI, 1.31–1.44) in 2016. The rate of detection of breast cancer in the ≥70 year age group was lower than that in the 40–69 year age group throughout the study period.

### 3.3. Comparison Age-Standardized Detection Rate with Age-Standardized Incidence Rate 

In the 40–69 year age group, the NHIS’s age-standardized detection rate of breast cancer increased annually from 106.1 in 2009 to 158.6 in 2015, and did not differ from the KCCR’s age-standardized incidence rate (Figure 3A).

In the ≥70 year age group, the NHIS’s age-standardized detection rate of breast cancer increased annually from 65.7 in 2009 to 120.3 in 2015, and was 1.9 to 2.7 fold of the KCCR’s age-standardized incidence rate of breast cancer (Figure 3B).

## 4. Discussion

The purpose of this study was to present the differences in epidemiological characteristics on a breast cancer screening program between the 40–69 year age group and the ≥70 year age group. The number of subjects and participants in breast cancer screening in the ≥70 year age group increased. This is probably due to the increased number of participants from the aging population and expanded average lifespan. 

Participation rates in breast cancer screening increased from 2009 to 2016 in both age groups, reflecting the ongoing publicity of the need and importance of cancer screening along with the need for public health [18]. In addition, the age group over 70 showed a lower participation rate than the 40–69 group, which is likely to be due to the lower susceptibility to, and severity of breast cancer in patients at the age of ≥70 [19].

The positivity rate in the 40–69 age group was higher than that in the ≥70 age group, which is likely to be due to the higher incidence of breast cancer in the former [17]. The positivity rate in the ≥70 age group increased annually during the study period. This suggests that the incidence of breast cancer among >70 aged women is increasing, in agreement with the Korean cancer statistics [20]. Also, the positivity rate in this study was much higher than the recall rate in Western countries [21]. The reason might be as follows: The high sensitivity of mammography as a screening method was emphasized so as not to miss the cancer cases in Korea while the threat of litigation against malpractice could cause a tendency of testing positive.

The PPV for breast cancer of the ≥70 age group increased over time and was higher than that of the 40–69 age group. The PPV was affected by the prevalence of breast cancer, sensitivity, and specificity of the screening tool [17]. The higher PPV in the age group of 70 and older indicates that breast cancer screening is still effective in women aged over 70. Since the breast density affects the accuracy of mammography [22], the PPV of the ≥70 age group with low breast density [23] might be high. The relatively lower PPV in the young group in this study could be explained by low prevalence of breast cancer and higher positivity rate of screening.

The detection rate of breast cancer per 1000 in the ≥70 year age group increased annually after 2009. This is similar to the case in the United Kingdom, where the number of women at the age of >73 diagnosed with breast cancer has increased consistently [24]. In addition, the age-standardized breast cancer detection rate was higher among women aged 40–69 than women aged 70 or older. However, in the age group of 70 or older, the NHIS’s age-standardized breast cancer detection rate was significantly higher than that of the KCCR’s age-standardized breast cancer incidence rate from 2009 to 2015. This means that breast cancer for the age group over 70 is likely to be diagnosed by a breast cancer screening program, not by self-examination or symptoms [19]. This is consistent with a report that screening is beneficial for older women for whom mammography has high sensitivity and specificity, and who have a high rate of detection along with PPV for breast cancer, low rates of retesting, and false positives [25]. Breast cancer has become a potentially curable disease with the development of new therapies over the past two decades [26]. Therefore, early detection is the most important way to avoid the development and progression of breast cancers regardless of age. 

To set the upper age limit for breast cancer screening, all-cause mortality, cancer specific mortality and stage shift are generally considered [6]. Although stage shifts or mortality rate were not evaluated in this study, it does show that the NHIS’s age-standardized breast cancer detection rate due to breast cancer screening was 1.9 to 2.7 fold of the KCCR’s age-standardized breast cancer incidence rate in the age group of 70 or older. Finally, the question of whether the upper age limit of screening should be increased, and to what level, is therefore of great importance in the drive to improve cancer outcomes in older women. One of the key arguments in favor of extending the upper age limit of screening is the increasing life expectancy of Western populations [21].

Because the data of this study used the whole data of breast cancer screening and the whole data of claims for breast cancer, the effect of breast cancer screening such as detection rate could be evaluated. However, there is no formal guideline for monitoring breast screening program performance in Korea. Especially, the interval cancer of breast cancer has increased 41.8 in 2009 to 58.5 in 2014 among 100,000 negative screening participants [17]. In addition to the researchers’ effort, the relevant indices about breast cancer screening have to be reported systemically and continuously by the NHIS who manage the two databases. There are many ways to detect breast cancer apart from the screening test, such as showing symptoms, regular check on exposing to risk factors like hormone treatment etc. However, we tried here to evaluate the effect of breast cancer screening only using a short detection time (six months) after screening. 

## 5. Conclusions

The comparison between the age-standardized detection rate of breast cancer from the data of NHIS and the age-standardized incidence rate of breast cancer from KCCR showed that the early detection effect of breast cancer screening was greater in women over 70 years old. Further studies on cost effectiveness, mortality rate, and stage shifts are needed to set the upper age limit for breast cancer screening.

## Figures and Tables

**Figure 1 healthcare-08-00132-f001:**
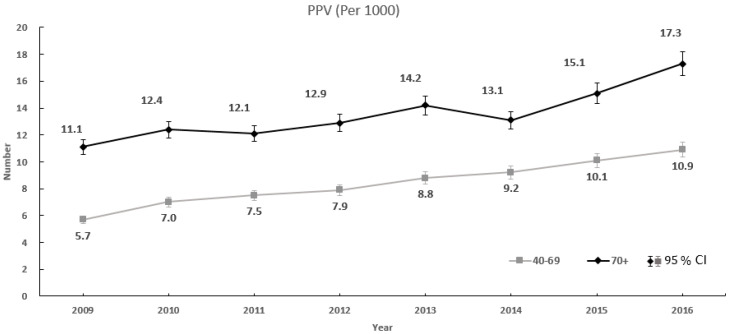
PPV of breast cancer screening by age group, 2009 to 2016.

**Figure 2 healthcare-08-00132-f002:**
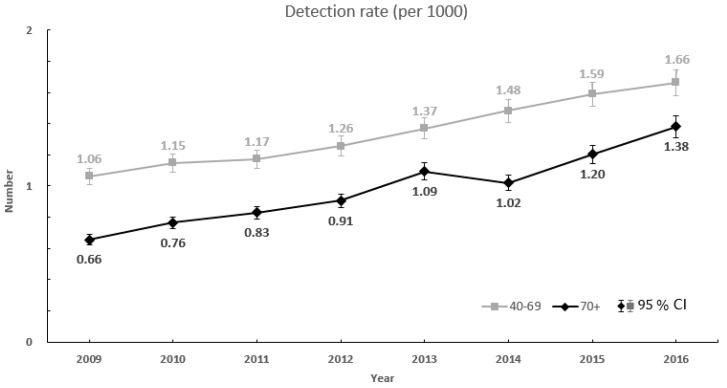
Detection rate of breast cancer by age group, 2009 to 2016.

**Figure 3 healthcare-08-00132-f003:**
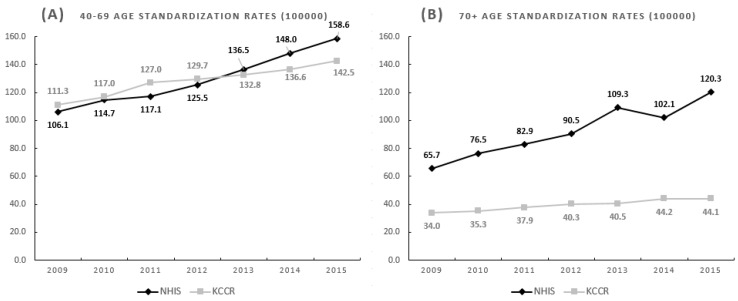
Age-standardized NHIS (detection rate) and KCCR (incidence rate) of breast cancer, 2009 to 2016. (**A**) 40–69 years of age, NHIS vs. KCCR; (**B**) ≥70 years of age, NHIS vs. KCCR. NHIS, National Health Insurance Service; KCCR, Korea Central Cancer Registry.

**Table 1 healthcare-08-00132-t001:** Numbers of subjects and participants and rates of participation and positivity, 2009 to 2016.

		2009	2010	2011	2012	2013	2014	2015	2016
**Subjects for Screening**	40–69	4,963,914	4,482,903	5,171,907	4,563,574	4,713,451	4,765,409	4,945,719	5,072,341
	≥70	948,517	918,220	1,129,966	1,012,101	1,049,574	1,083,725	1,111,346	1,175,298
	Total	5,912,431	5,401,123	6,301,873	5,575,675	5,763,025	5,849,134	6,057,065	6,247,639
**Participants**	40–69	2,597,693	2,525,167	2,909,618	2,858,449	2,918,093	3,031,223	3,222,881	3,432,426
	≥70	377,662	381,887	458,570	483,923	497,824	538,924	551,969	606,112
	Total	2,975,355	2,907,054	3,368,188	3,342,372	3,415,917	3,570,147	3,774,850	4,038,538
**Participation** **Rate (%)**	40–69	52.3	56.3	56.3	62.6	61.9	63.6	65.2	67.7
	≥70	39.8	41.6	40.6	47.8	47.4	49.7	49.7	51.6
	Total	50.3	53.8	53.4	59.9	59.3	61	62.3	64.6
**Positivity** **Rate (%)**	40–69	18.7	16.3	15.7	15.9	15.6	16.1	15.7	15.2
	≥70	5.9	6.2	6.8	7.0	7.7	7.8	8.0	8.0
	Total	17.0	15.0	14.5	14.6	14.5	14.9	14.6	14.2

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
