# Peer review of "Age-Standardized Breast Cancer Detection Rates of Breast Cancer Screening Program by Age Group in Korea; Comparison with Age-Standardized Incidence Rates from the Korea Central Cancer Registry"

_healthcare, 2020, doi:10.3390/healthcare8020132_

Round 1

Reviewer 1 Report

Apparently the authors aimed at collecting more evidence to support/refute the current Korean national recommendation of screening women above 70 years of age. Selection of upper limit of target age for screening depends on several factors - RCT showing mortality reduction, risk of over diagnosis, co-morbidities, cost-effectiveness, prioritisation of resources etc. Just because screening tests detect more case and has better PPV may not be the right justification. Interestingly, the age-specific incidence of breast cancer in South Korean women peaks quite early - around 50 years of age. 

The screening data shows the extensive screening activities ongoing in the country. The very high participation rate (>100% for a round of screening that lasts for 2 years) indicates that the reported number of screening includes women being screened out of turn and/or the diagnostic mammographies. An impressive number of women >69 yrs are already undergoing screening.

The very high screen positivity in the 40-69 yr age group is surprising, specially in a population that is heavily screened. Most of the mammography screening programmes in Europe have screen positivity between 5-8% even though the breast cancer incidence in these countries is much higher than that in SK. PPV is better presented as a percentage and depends on the disease prevalence. The low PPV (~1% in 40-69 yr vs. 10-15% in Europe) may still be explained by the low disease prevalence but low prevalence does not explain the low PPV in younger age group. 

The detection rate is normally presented as /1000 screened is again much lower than that reported in European programmes and may be explained by the lower disease incidence in SK. Higher rate of detection in the older age compared to that reported in the PBCR is possibly because of relatively recent onset of screening in this age group.

The article can have better scientific value if the authors focus on the quality issues of the programme, quality of data etc. The linkage between the screening and the insurance database may allow the authors to look at the interval cancers - a critical quality indicator of screening.

Reviewer 2 Report

The aim of this interesting paper is to compare the epidemiological characteristics on breast cancer screening program between 40-69 years of age and ≥ 70 years of age. The Screening program started in Korea in 1999, modified in 2009 reccomending for women over 69 to selectively perform breast cancer screening using mammography in consideration of individual risk and evaluation, as reported in 2015 by other Authors (ref 8)

Authors calculated the age-standardized detection rate of breast cancer screening program (Jan 2009 to Dec 2016) and compared it with age-standardized incidence rate from South Korea Central Cancer Registry (Jan 2006 to August 2017)

Question: How was the individual risk and evaluation made for women aged >69 years?

I suggest to add in introduction the paper of Lee JH et al reporting data from 1993 and 2002

(Lee JH, Yim SH, Won YJ, et al. Population-based breast cancer statistics in Korea during 1993-2002: incidence, mortality, and survival. J Korean Med Sci 2007;22 Suppl:S11 – 6)

Furtehrmore, the life span of women over 60 Yrs in Korea could be added in Introduction . In Western Countries it is around 17 years

In discussion, Authors could cite after the good paper of UK (ref 20) cite also the excellent long term survival in breast cancer patients over 69 of the GRETA (G.Mustacchi et al. Future Oncology 2015, 11 (6),933-941)

Author Response

Response to reviewer’s comments

  • We really appreciate you for your time and effort on our manuscript. During the revision, we recognized the weakness of the manuscript, and could improve its quality. We agree with the issues that you pointed out and tried to reflect them in the revised manuscript. We hope our effort would be satisfactory to you.
  • Authors calculated the age-standardized detection rate of breast cancer screening program (Jan 2009 to Dec 2016) and compared it with age-standardized incidence rate from South Korea Central Cancer Registry (Jan 2006 to August 2017)

      -> Thanks for your comments.

  • Comment 1: How was the individual risk and evaluation made for women aged >69 years?

  • Response: The elderly’s individual risk and its evaluation for breast cancer would be made at the doctor’s office based on the elderly’s medical record (previous mammogram or physical examination etc) and her past histories. We did not reflect this part in text because it is beyond this study.

  • Comment2: I suggest to add in introduction the paper of Lee JH et al reporting data from 1993 and 2002 (Lee JH, Yim SH, Won YJ, et al. Population-based breast cancer statistics in Korea during 1993-2002: incidence, mortality, and survival. J Korean Med Sci 2007;22 Suppl:S11 – 6)

  • Response: Your suggestion sounds very good. Considering the entire context, we added the incidence rate and mortality rate in 1993 and one sentence in introduction (lines 44-46).(in red)

  • Comment 3: Furthermore, the life span of women over 60 Yrs in Korea could be added in Introduction. In Western Countries it is around 17 years

  • Response: You have raised an important point. But, since our paper did not deal with mortality, we made a decision we did not reflect your opinion here. We are sorry about that.

  • Comment 4: In discussion, Authors could cite after the good paper of UK (ref 20) cite also the excellent long term survival in breast cancer patients over 69 of the GRETA (G.Mustacchi et al. Future Oncology 2015, 11 (6),933-941) Update of the Phase III trial ‘GRETA’ of surgery and tamoxifen versus tamoxifen alone for early breast cancer in elderly women.

  • Response: After reviewing two papers you mentioned, we tried to reflect some contents as below (line 233-236), but we did not reflect the ‘GRETA’ because we thought it is beyond the contents of this paper.(in red)

  • Thank you. We have revised all the things you pointed out that we can do. And, we will try to reflect on the things that I couldn't fix this time in our next study.

Round 2

Reviewer 1 Report

The manuscript may be accepted